# Fungal Development and Callose Deposition in Compatible and Incompatible Interactions in Melon Infected with Powdery Mildew

**DOI:** 10.3390/pathogens10070873

**Published:** 2021-07-10

**Authors:** Paola Beraldo-Hoischen, Caroline Hoefle, Ana I. López-Sesé

**Affiliations:** 1Instituto de Hortofruticultura Subtropical y Mediterránea “La Mayora”, Consejo Superior de Investigaciones Científicas-Universidad de Málaga (IHSM-CSIC-UMA), Estación Experimental “La Mayora”, Avda. Dr. Wienberg, s/n, E-29750 Algarrobo-Costa, Málaga, Spain; paola@eelm.csic.es; 2Center of Life and Food Science Weihenstephan, Technische Universität München, Emil-Ramann Strasse 2, 85350 Freising-Weihenstephan, Germany; c.hoefle@wzw.tum.de

**Keywords:** cucurbits, disease, fungal growth, *Podosphaera*, resistant, temperature

## Abstract

Two post-haustorial resistance mechanisms (types I and II) against powdery mildew, caused by *Podosphaera xanthii*, have been described previously in melon according to the arresting of fungal development and the timing of hypersensitive response (HR) in host cells. In our work, host-pathogen interactions between races 1, 2, and 5 of *Podosphaera* and several melon accessions carrying different resistance genes, have been characterized by observing several parameters, such as the number of fungal penetration points with callose accumulation, the number of epidermal cells with callose accumulation in their cell walls, and the number of conidiophores developed. Influence of temperature was observed in some cases affecting the timing of fungal development arrest. According to our results, besides the compatible interaction, four different resistance behaviors in the plant–pathogen interaction have been observed herein: type I and II, as described previously, as well as an earlier and a later type II: IIa and IIb, respectively. Melon genotypes showing post-haustorial resistance mechanism types IIa and IIb against powdery mildew, seem to show different behavior according to temperature, affecting fungal development, mainly those genotypes carrying QTL of linkage group V for powdery mildew resistance, such as “TGR-1551”.

## 1. Introduction

The fungal disease known as powdery mildew is a limiting factor for the production of melon throughout the world [1]. *Golovinomyces cichoracearum* [2] and *Podosphaera xanthii* [3], are the two main causal agents of this disease. *Podosphaera xanthii* occurs more frequently in subtropical and tropical areas [4] and, in coastal zones of the Spanish provinces of Malaga and Almeria, this species has been identified as the sole cause of powdery mildew of cucurbits [5,6,7]. Many races of *P. xanthii* have been identified so far based on the differential response of resistance/susceptibility of a set of melon genotypes [8,9]. Disease resistance is an important objective of melon-breeding programs, which implies the knowledge of the inheritance of the resistance and of the mechanisms involved in it.

Plants have evolved sophisticated and efficient mechanisms to prevent the invasion of their tissues by pathogens, and disease rarely occurs. They defend themselves against fungi by mechanisms expressed either constitutively, before pathogen attack, or induced, after fungus recognition, which provokes the activation of a vast array of chemical and physiological changes [10]. Quantitative cytological recordings of incompatible interactions have revealed distinct putative host–cell defense mechanisms conferring arrest of fungal development at different specific stages [11,12,13]. Two host–cell resistance mechanisms have been detected: cell wall reinforcement at sites of attempted penetration (effective papilla), and an active, rapid death of attacked epidermal cells, named hypersensitive response (HR), which can be visualized by whole-cell autofluorescence under UV excitation [12,14,15]. HR is associated with several processes such as rapid generation of reactive oxygen species (ROS), direct ion flux through the plasma membrane, synthesis of antimicrobial compounds (phytoalexins), strengthening of plant cell via cross linking of cell wall polymers (callose and lignine) [16,17], and transcription of defense-related genes (pathogenic-related proteins, PR).

Regarding powdery mildew diseases in general, various mechanisms of resistance have been described so far, and can be roughly classified as pre- and post-haustorial resistances [18,19,20]. Pre-haustorial resistance is characterized by the presence of effective papillae under the fungus penetration attempt, which blocks fungal development. This resistance has been observed in barley genotypes containing the *mlo*-resistant gene against *Blumeria graminis* [21]. It is a race non-specific resistance, and it does not damage the cell. Pre-haustorial resistance has not been observed in melon so far. Post-haustorial resistance mechanisms have been described in this species, though. Concretely, regarding powdery mildew in melon, Kuzuya et al. [20] distinguish two different post-haustorial resistance behaviors, type I and type II, which stop fungal development at different stages depending on the timing of HR in the epidermal cells. In type I, fungal growth is arrested at one or two primary hyphae stages, around 48 h post-inoculation (hpi), showing few cells with callose accumulation in the penetration points and in their cell walls. In type II, HR in the epidermal cells occurs later in time and a larger fungal development can be observed in a higher number of cells, showing callose accumulation around penetration points and in cell walls. Some conidiophores can also be seen at 120 hpi. In a compatible reaction, conidia germinate within 12 h post inoculation, an haustorium is formed within 24 h, and germ tube branching and hyphae elongation began within 48 h. New conidiophores were initiated within 120 h, which matured within 240 h [20].

The principal aim of this report is the observation, at microscopic level, of the host–pathogen compatible and incompatible interactions between races 1, 2, and 5 of *Podosphaera xanthii* and several melon accessions carrying different resistance genes, including two recombinant inbreed lines (RIL) obtained from the cross between “TGR-1551” (resistant to *P. xanthii* races 1, 2, and 5) and “Bola de Oro” (susceptible to these three races). The possible association between differential resistance to pathogen and the presence or absence of putative resistance genes are discussed.

## 2. Results

### 2.1. Conidiophores Counting at 120 hpi

In order to observe the fungus development, the number of mature conidiophores was counted in the different melon accessions grown at TUM at 22 °C, 70% RH, and 16/8 light/dark (Figure 1). The highest number of mature conidiophores, more than four per conidium, was detected for “Bola de Oro” inoculated with races 1, 2, and 5, “PMR 45” inoculated with races 2 and 5, and “WMR 29” with race 5 (Figure 1). In all these cases, the response to powdery mildew was susceptible (compatible interaction). The lowest values, approximately one or none conidiophore per conidium, were identified in cases of resistant response (incompatible interaction), concretely in “PI 414723” for races 2 and 5, NIL21, RIL88, and RIL276 for races 1, 2, and 5, in “PMR 45” for race 1, and in “WMR 29” for races 1 and 2. The resistant melon line “TGR-1551” showed, however, intermediate scores for the three races (Figure 1).

According to the results from the ANOVA test for the number of conidiophores counted at 120 hpi for the three races tested, three different groups of accessions could be observed based on their response: one with values around zero (incompatible interaction), a second group with values above three (compatible interaction), and a third one with intermediate scores and corresponding to “TGR-1551”.

### 2.2. Callose Accumulation in Penetration Points and Cell Walls at 48 hpi

In order to estimate the fungal penetration attempts in the different melon accessions, as well, as the plant response to them, callose accumulation around the penetration attempts as well as in the cell walls, were observed during fungal development (Figure 2). These observations were carried out at TUM at 22 °C, 70% RH, and 16/8 light/dark. In general terms, no significant differences among the response to the three powdery mildew races were detected for the number of penetration points with callose accumulation in “TGR-1551”, “Bola de Oro”, “PI 414723”, NIL21, and RIL276 (Figure 2), with an average value of around 1.7–2 points per conidium for the three powdery mildew races. These melon lines have been described as showing a similar behavior to the three races evaluated: all of them are resistant to the three powdery mildew races, except “Bola de Oro”, which is susceptible. Differences among the resistance to the three races evaluated were observed in the accessions “PMR 45” and “WMR 29”. The accession “PMR 45” showed the lowest value (around 1) for race 1, as expected for its described incompatible interaction. In the already described compatible interaction of “WMR 29” and race 5, the highest values were detected (Figure 2). The mean values observed in the recombinant inbred line RIL88, coming from “TGR-1551”, were also different for the three races. The lowest value was noticed when inoculated with race 1 (1.20). Regarding powdery mildew race, significant differences were found among lines for each race tested. In general, in the compatible interactions of “Bola de Oro” (races 1, 2 and 5), “PMR 45” (races 2 and 5), and “WMR 29” (race 5), more than two penetration points with callose accumulation per conidium were observed, the values below two being those one detected for incompatible interactions, such as those observed in “PMR 45” (race 1), “WMR 29” (races 1 and 2), and NIL21 (races 1, 2, and 5). This was not detected, however, in “PI 414723” (races 2 and 5, incompatible interaction) and in “TGR-1551” (races 1, 2 and 5), which showed more than two penetration points per conidium, and were not significantly different from each other and from the susceptible line “Bola de Oro” at that time. The recombinant inbred lines RIL276 and RIL88 displayed above two penetration points per conidium in two out of the three races (1 and 5, and 2 and 5, respectively). Based on these results, the counting of penetration points with callose at 48 hpi and 22 °C seems not to be an adequate parameter to use in order to discriminate clearly between compatible and incompatible interactions, possibly due to that 48 hpi is a very early stage in powdery mildew development to observe differences.

Regarding callose accumulation in cells walls, significant differences were found among races and also among lines for the same race tested (Figure 3). Significant differences among powdery mildew races 1, 2, and 5 were observed in “Bola de Oro” (1, 0.44, 0.78), “PI 414723” (1.3 and 1 for races 2 and 5, respectively; race 1 was not tested), “WMR 29” (1.3, 1.2, 0.8) (lowest value for race 5), RIL88 (0.96, 1.5, 1.94) and RIL276 (1.5, 1, 0.7). No differences were noticed for “TGR-1551” (1, 0.7, 1), NIL21 (around 1.3 for the three races) and “PMR 45” (0.7, 0.91, 0.92) (Figure 3). The lack of differences detected in “PMR 45” (resistant to race 1, susceptible to races 2 and 5) could be due to the non-presence or short development of the fungus in many of the conidia observed for race 1 (value of 0.7, incompatible interaction). Moreover, no differences were noticed between “Bola de Oro” and “TGR-1551” for each of the three races tested. In general, highest values (value above 1) were detected in most of the described incompatible interactions, although observations at later time points post inoculation could be needed to discern properly between compatible and incompatible interactions.

According to all these results, none of the parameters evaluated at 48 hpi and 22 °C (number of penetration points, presence of callose in cell walls, number of primary hyphae, and the ratio penetration points/cells with callose) showed significant differences between the resistant melon line “TGR-1551” and the susceptible “Bola de Oro”, both parental lines of the NIL and RIL used (Figure 4). Likewise, the number of conidiophores observed at 120 hpi in “TGR-1551” was intermediate between the values observed for the compatible and incompatible interactions. Therefore, it is not clear if the interaction between “TGR-1551” and *P. xanthii* could be described as a compatible or an incompatible interaction, as estimated by these parameters under the environmental conditions tested.

In order to clarify and understand the resistance of “TGR-1551” to powdery mildew, new observations were carried out at different time points, since higher time points could be more informative.

### 2.3. Fungal Development at 72 hpi and 96 hpi and Conidiophores Observation at 240 hpi (Conidia Mature Stage)

Three plants from “Bola de Oro”, “TGR-1551”, and NIL21 (resistant genotype carrying QTL *Pm-R*, coming from the parental line “TGR-1551”) were inoculated with races 1, 2, and 5, and samples were taken at 72 in order to estimate fungal development, following the methods explained before. Samples from “Bola de Oro”, “TGR-1551” were also taken at 96 hpi. These observations were carried out at TUM at 22 °C, 70% RH, and 16/8 light/dark.

In general terms, and according to the number of penetration points with callose and the ratio P. points/callose observed that at 72 hpi, the three races tested of *P. xanthii* reached the highest development in “Bola de Oro” (compatible interaction) and the lowest one in NIL21 (incompatible interaction) (Figure 5). “TGR-1551” presented intermediate values for these parameters. The lowest fungus development might be related with the presence of a resistant response which could be traduced in a higher number of plant cells with callose in the resistant melon genotypes. As a matter of fact, the highest values were found for NIL21, the lowest one for “Bola de Oro”, and intermediate values for “TGR-1551” (Figure 5).

Regarding the statistical analyses, significant differences were detected among the three accessions for each of the three races tested for the number of penetration points, as well as for the ratio P. points/callose (Figure 5). Differences were found among the three genotypes for the number of cells with callose although only for races 1 and 2, since NIL21 and “TGR-1551” presented similar values for race 5. Furthermore, the differences were observed for the number of hyphae for races 2 and 5, since “TGR-1551” and “Bola de Oro” showed similar values for race 1 (Figure 5). Considering the differences among races for the same genotype and parameter, they were found for the number of penetration points and P.points/callose in “TGR-1551” and NIL21, while in “Bola de Oro” the values of these two parameters were different only when inoculated with race 5. The three genotypes reached the highest scores for penetration points and P.points/callose. “TGR-1551” and NIL21 also showed the highest scores for the number of cells with callose, when inoculated with race 5. “Bola de Oro” showed no differences among races for the number of hyphae and dead cells.

For the observations at 96 hpi, three plants per genotype (“TGR-1551” and “Bola de Oro”) were inoculated only with race 1 (Figure 6), since no large differences were observed previously among races for these parameters. In this case, significant differences were detected between “TGR-1551” and “Bola de Oro” for all the variables tested, except for the number of primary hyphae. The highest differences were found for the number of cells with callose and in the ratio P. points/callose (Figure 6). We observed around 50% more cells with callose accumulation in their cell walls in “TGR-1551” than in “Bola de Oro”; in addition, the number of P. points was quite similar, a 15% more in “Bola de Oro” than in “TGR-1551”.

We could conclude that “TGR-1551” is significantly different from “Bola de Oro” at time points above 72 hpi for the number of cells with callose, as well as for the ratio established between the number of penetration points and number of cells with callose.

Since conidiophores were already observed and counted at 120 hpi in the resistant melon line “TGR-1551”, new observations were considered in plants inoculated with race 1 at 240 hpi for mature conidia stage, according to Kuzuya et al. [20]. In that case, we detected some conidiophores in “TGR-1551” at 240 hpi, although the number was quite low in comparison to “Bola de Oro” (Figure 7), where the count was non-viable due to the huge amount of them at that time point. In previous inoculations carried out in our team at La Mayora (Spain), “TGR-1551” had never shown such a development of the fungus, therefore another variable might be influencing this fungal development in “TGR-1551”. Concretely, the average day temperature registered during those previous experiments at La Mayora in climatic chambers in different seasons were usually higher than 25–30 °C, and “TGR-1551” showed a clear resistance to powdery mildew. Only some slight and brief fungal growth was observed occasionally in some experiments carried out in autumn. Therefore, that variable could be likely be temperature, since in the literature, some melon genotypes have been described as showing different responses to powdery mildew depending on the temperature, such as ANC-57 [22], and “Quincy”, ENN2, EMN2, and HN21 [23]. In order to estimate if temperature employed during the experiments (22 °C) at TUM (Germany) could have been influencing the fungal development in “TGR-1551”, an additional experiment with different temperatures was carried out.

### 2.4. Fungal Development at Different Temperatures

Plants from several melon accessions were used in order to observe fungal development at different temperatures and at different times post inoculation. Callose accumulation in penetration points and cell walls, as well as the number of primary hyphae, were evaluated at 48, 72, 96, and 120 hpi, the number of conidiophores being also recorded at 120 hpi. Three plants per melon accession and powdery mildew race were used in all cases.

No significant differences were observed in “TGR-1551” and “Bola de Oro” at 72 hpi for the evaluated parameters between the experiments carried out in both places, TUM (Germany) and “La Mayora” (Spain), at low temperature (22 °C and 18–22 °C, respectively) (data not shown). We did find, however, significant differences between both temperatures in the experiments carried out at “La Mayora” for all the parameters tested for the same genotype and time after inoculation for all the evaluated lines, except for the number of hyphae in “Bola de Oro” at 48, 96, and 120 hpi, and in “PMR 45” at 120 hpi for the number of hyphae and conidiophores (data not shown). Likewise, we did not find differences for the number of penetration points with callose accumulation in “TGR-1551” at 48 hpi (Figure 8B) and for the number of cell walls with callose at 72 hpi. The number of penetration points with callose increased along the time in all the tested lines inoculated with race 2 for both temperatures, except for “WMR 29” (resistant to race 2) where the fungal development stopped mostly at 48 hpi, showing a maximum of one penetration point per conidium approximately (Figure 8A). The highest values of this parameter for both temperatures were observed, as expected, at 120 hpi in the lines susceptible to race 2 (“Bola de Oro”, “PMR 45”), and in “TGR-1551” at low temperature (Figure 8B).

Regarding the number of penetration points with callose observed in each melon line at different temperatures, “WMR 29”and NIL21 (resistant to race 2 and carrying QTL *Pm-R*) at 48 hpi and 120 hpi, “PMR 45” at 48 hpi, and “TGR-1551” at 72, 96, and 120 hpi, showed larger scores at low temperature than at high temperature. By contrary, “PMR 45” at 120 hpi and “Bola de Oro” at 48, 72, 96, and 120 hpi exhibited larger scores at high temperature than at low one (Figure 8A,B). This parameter, number of penetration points with callose, might be useful to describe the fungal development, since it presented the highest differences between the resistant and susceptible accessions at both temperatures examined. The ratio penetration points and cells with callose would likely make this difference more remarkable.

Regarding the number of conidiophores at 120 hpi, the susceptible melon lines, “Bola de Oro” and “PMR 45”, showed a higher number at high temperature than at low. The resistant genotype “TGR-1551” showed the higher value at low temperature. The genotype “WMR 29” did not present any conidiophores in any case. The number of conidiophores counted at 120 hpi was significantly different between “TGR-1551” and “Bola de Oro” at both temperatures. At high temperature, “TGR-1551” did not show any conidiophores while at low temperature it did show some. “Bola de Oro” showed nearly the double of conidiophores per conidium (data no shown) at high temperature than at low temperature.

In summary, differences between “TGR-1551” and “Bola de Oro” could be observed at high temperatures at all-time points tested for the number of penetration points with callose (Figure 8B,C), number of cells with callose and primary hyphae, and for conidiophores counted at 120 hpi (data not shown), due to the no growth of most of the conidia on “TGR-1551” at that temperature. Nevertheless, at low temperatures, as described before, differences among these lines started to be noticed at 72 hpi for the number of cells with callose and above 120 hpi for the number of conidiophores. 

As conclusion, it could be considered that the set of these three parameters, number of conidiophores at 120 hpi and on, number of penetrations points, and number of cells with callose after 72 hpi, might discriminate between compatible and incompatible interactions, as well as among different resistance responses. 

## 3. Discussion

The results obtained in this report confirmed the differences between compatible and incompatible interactions observed in plant-powdery mildew system, as described previously for melon and other plant species affected by pathogens causing powdery mildew [20,24,25,26].

In compatible interactions, fungal development is not stopped by any host defense mechanism of resistance. In incompatible host–pathogen interactions, where arresting of fungal development occurs, several types of behaviors have been described according to the mechanisms involved. Two general types have been described so far in different species [12,15,18,19,20,27]. One is called pre-haustorial, where fungal arrest takes place before the haustorium formation, and the second one is known as post-haustorial, where the haustorium is formed complete or incompletely, allowing fungal development. The pre-haustorial, which has not been observed in melon yet, has been described to be a non- race-specific response [28,29,30], and it is characterized by the presence of an effective papillae under most penetration attempts, which blocks fungal penetration, and fungal development is arrested at germ tube stage [27]. On the other hand, in the post-haustorial, which has been described as being a race-specific response, two different powdery mildew-melon resistance behaviors can be distinguished, type I and II [20], according to the timing of the presence of HR in the epidermal cells affected. In other species, Hückelhoven et al. [27] distinguished two different post-haustorial resistance responses: one which takes place in the epidermal cells, similar to that one described by Kuzuya et al. [20] as type I in melon, and a second one, which affects the mesophyll cells under the penetrated epidermal cells, restricting nutrient supply in the penetrated epidermal cells. The mechanisms occurring in the second post-haustorial resistance described by Hückelhoven et al. [27] might be similar to those which take place in melon for the type II described by Kuzuya et al. [20], where the HR response occurs later. Therefore, in general terms, in type I the fungal development stops at germ tube or at one to two hyphae stages and in type II, a further fungal development is noticed, with secondary hyphae and even some conidiophores [18,20].

Diverse variables have been used to characterize the different responses observed in the melon-powdery mildew interaction. In this work, several main parameters were used: the number of fungal penetration points with callose accumulation, the number of epidermal cells with callose accumulation in their cell walls, and the number of conidiophores developed. The number of primary hyphae and the estimated ratio penetration points/cells with callose were also considered in most cases. According to our results, besides the compatible interaction, four different resistance behaviors in the plant–pathogen interaction have been observed: type I and II, as described by Kuzuya et al. [20], as well as an earlier and a later type II, IIa, and IIb respectively.

The number of penetration points, detected as yellow rings by aniline staining, increased along the time in compatible interactions [20,24,31], and haustorium might be well developed since the fungus has not stopped its development [20]. From our results, we cannot conclude if the fungus was able to develop an haustorium, although a well-developed haustorium may be needed in order to reach a higher development level such as the one observed in compatible interactions.

In the resistance response type I, the number of penetration points showed the lowest value among the different types of behaviors observed herein, and fungal development was arrested at early stages (48 hpi). A low number of penetration points and epidermal cells with callose were observed. Normally, under every penetration point, a cell with callose accumulation in their cell wall was detected, showing a ratio penetration points/cells with callose of one, approximately. The number of primary hyphae was around 1 or 2, and no conidiophores were observed at 120 hpi. We detected this response in “TGR-1551”, “WMR 29”, and “NIL21” inoculated with race 2 at high temperature (race 2 was the only one used at high temperature), and in “PMR 45”, and “WMR 29” inoculated with race 1 at low temperature. Kuzuya et al. [20] noticed this same resistance response in “PMR 45” and “WMR 29” inoculated with race 1, as well as they also observed it in “WMR 29” inoculated with race 2. According to our results, “WMR 29” inoculated with race 2 presented a response type I only at high temperatures, since at low temperature, the fungal development was not stopped at first germ tube stage. At low temperature we could observe one or two primary hyphae in some conidia, and even some conidiophores at 120 hpi (12 from 90 conidia in one out of three leaves). Such differences have been also described by other authors. For instance, “PI 124112” was considered as showing resistance type I when inoculated with race 2 [20], although Cohen and Eyal [32] observed advanced development stages too. 

Another resistance response observed in our study, considered as type II, is characterized by a number of penetration points and epidermal cells with cell walls with callose higher than in the resistance considered type I. Moreover, the ratio penetration points/cells with callose is higher than one. The number of penetration points at 48 hpi is not significantly different from the observed in the compatible interactions, even for higher time points (>48 hpi). The number of primary hyphae can reach a value of three, like in compatible interactions, and none or only a few conidiophores were detected at 120 hpi. This resistance has been clearly observed in our study in “PI 414723” with *P. xanthii* races 2 and 5 (race 1 was not tested with this genotype), as described by Kuzuya et al. [20], as well as for RIL88 and RIL276 (at low temperature, not tested at high), inoculated with the three races. A third resistance behavior, similar to type II but with a faster fungal development, named herein as type IIa, was observed in “WMR 29” inoculated with race 2 and in NIL21 with the three races also at low temperature. The fungal development was arrested later than in type I, and nearly one penetration point per cell with callose has been observed. A fourth resistance behavior, slower than type II, named IIb, was observed in “TGR-1551” at low temperature inoculated with the three races, showing an intermediate score between the susceptible and resistant lines for the number of conidiophores. At high temperature “TGR-1551” presented a clear resistance to powdery mildew. The results obtained in this work showed that temperature could influence the fungal development in “TGR-1551”, NIL21, and “WMR 29”. As we mentioned before, other authors have previously described temperature as an environmental factor which could be affecting the resistance of several melon genotypes to powdery mildew [22]. All these melon lines carry a QTL or a gene in linkage group V for powdery mildew resistance: “TGR-1551”, NIL21 (resistant to races 1, 2, and 5 due possibly to the dominant gene, *Pm-R*, in LG V) and “WMR 29” [resistant to races 1, 2, and 3, and carrying one dominant gene *Pm-w* or *Pm-B* [33,34]. According to this information we could suppose that, either the same genes controlling powdery mildew resistance in LG V, or some gene located close to these resistant genes could be significantly affected by temperature. 

Other specific temperatures experiments would be necessary in order to verify these results, and identify which mechanisms are involved in these processes. In any case, some associations could be established between the genes involved in the powdery mildew resistance in the different melon genotypes tested, and the phenotype of the different plant-pathogen interactions described herein. The resistance type I was observed in the melon lines “WMR 29”, NIL21, and “TGR-1551” inoculated with race 2 at high temperature, and in “WMR 29” inoculated with race 1 at low temperature. These three melon genotypes have at least one gene or QTL controlling powdery mildew resistance which is located in linkage group V [33,35,36]. Kuzuya et al. [20] also observed this resistance in “WMR 29” inoculated with races 1 and 2, as well as in other accessions with a dominant gene or QTL in LG V, such as “Edisto 47” [37] inoculated with race 1, and in “PI 124112” [38] inoculated with races 1 and 2. The melon line “PMR 45”, inoculated with race 1, showed also the same resistance in our work and in Kuzuya et al. [20]. Its resistance gene, *Pm-1*, was identified in LG IX [39]. Other genotypes with gene *Pm-1*, such as “PMR 5” and “PMR 6” [40], were also considered to show a resistance type I [20,41]. It seems that genotypes, carrying a gene or QTL-controlling powdery mildew resistance in LG V or IX, show a resistance type I. On the other hand, genotypes with genes or QTL located in LG II or XII seem to have a resistance type II. The melon line “PI 414723”, with a resistance gene located in LG II controlling at least resistance to races 1, 2, 3, and 5 [33,35,42], showed a resistance type II according to our results, as well as it is described by Kuzuya et al. [20]. This same resistance was noticed in RIL88 and RIL276, both resistant to powdery mildew due to the presence of a possible recessive gene in LG XII coming from “TGR-1551” [43]. The melon line “PI 124112”, with a QTL in LG XII which confers resistance to race 5 [38], also showed a resistance type II [20].

Concluding, according to our results, besides the compatible interaction, four different resistance behaviors have been observed in the plant–pathogen interaction: type I and II, as described by Kuzuya et al. [20], as well as an earlier and a later type II, IIa, and IIb respectively. These types IIa and IIb are mainly shown by melon genotypes having several genes involved in their resistance to powdery mildew, in which, epistasis issues could have taken place among them, being likely responsible of the differential response to temperature.

In further studies and using the information obtained after the sequencing of the melon genome, it would be interesting to saturate the genomic regions likely involved in the resistance to powdery mildew conferred by “TGR-1551” by using SNP, as well as to determine possible candidate genes associated with this resistance.

## 4. Material and Methods

In order to observe the host–pathogen compatible and incompatible interactions between *Podosphaera xanthii* and *Cucumis melo* L., three powdery mildew races (1, 2, and 5) as well as several melon accessions carrying different resistance genes were tested. Three plants were used per melon accession and powdery mildew race tested. The melon accessions included the parental lines “TGR-1551” and “Bola de Oro”. The genotype “TGR-1551” has been described as resistant to races 1, 2, and 5 of powdery mildew. This resistance is conferred by one dominant and one recessive gene and it is described as a double dominant recessive epitasis [44]. The genotype “Bola de Oro” is a Spanish cultivar susceptible to these three powdery mildew races. From both parental melon accessions, three lines were selected by phenotypic and molecular analyses from previous studies: the RIL88 and RIL276, two recombinant inbred lines obtained after seven self-crossings from the cross” TGR-1551” × “Bola de Oro”, both also resistant to races 1, 2 and 5, due possibly to the presence of a recessive gene from “TGR-1551” [43]; and the NIL21, a near-isogenic line obtained after five backcrossing from the same cross “TGR-1551” × “Bola de Oro” and resistant to races 1, 2, and 5 possibly due to the dominant gene related to the QTL *Pm-R* [36]. Moreover, several melon accessions with differential response to powdery mildew races were used: “PMR 45”, resistant to race 1 and carrying the dominant gene *Pm-1* or *Pm-A* [34,45], “WMR 29”, resistant to races 1, 2, and 3 and carrying one dominant gene *Pm-w* or *Pm-B* [33,34], and “PI 414723”, resistant to races 1, 2, 3, and 5 and conferred by different genes according to various authors [33,35,42,46]. The experiments with all these melon lines were carried out in two locations, and plants were grown in two different environments: at 22 °C, 70% RH, 16/8 h light/dark in a growth chamber in the department of Phytopathology at the Technische Universität München (TUM), in Germany, and at 18–22 °C and 25–30 °C in a climatic chamber at the Experimental Station of the IHSM “La Mayora” (CSIC-UMA) in Malaga, Spain. 

The inoculations were carried out on the second true leaf of each plant using an inoculation tower. The *P. xanthii* isolates employed for the experiments were “27” (Race 1), “2204” (Race 2), and “C8 Cris” (Race 5). Race identification of these fungal isolates was based on the reaction of a set of differential melon lines to each isolate, as described by Bardin et al. [9]. Conidia were obtained from monosporic culture which were kept in cotyledons from the susceptible melon accession “Bola de Oro”, and from zucchini squash placed on Petri disks with Bertrand medium [47] under axenic conditions. The conidia were subcultured on new cotyledons every two weeks approximately.

Several parameters have been taken into account based on Kuzuya et al. [20]. Thus, the number of conidiophores (new conidia initiation stage) at 120 h post-inoculation (hpi), penetration points and cell walls with callose accumulation, both at 48 hpi (when the highest differences detected among susceptible and resistant accessions were observed), and number of primary hyphae, were recorded in “PMR 45”, “WMR 29”, “PI 414723”, “TGR-1551”, “Bola de Oro”, NIL21, RIL88, and RIL276 for races 1, 2, and 5 of *P. xanthii*. Inoculated leaves were cut in half and kept in tubes with a dilution ethanol–acetic acid 7:1, and employed for fungal development observation, conidiophore counting and callose accumulation detection. 

### 4.1. Spore Germination and Number of Conidiophores

In order to observe conidia, hyphae, and conidiophores, half leaf of each inoculated plant was taken and stained with an ink/acetic acid solution (incubation in 10% blue ink Pelikan within 25% acetic acid for 1 min and then washed) and visualized under light microscope (5×, 10×, 20×). The number of conidiophores was counted at 120 hpi in 30 conidia per sample.

### 4.2. Callose Accumulation in Penetration Points and Cell Walls for HR Estimation

Callose accumulation in penetration points and in cell walls was detected with the double aniline-blue (aniline blue 0.01% in 7 mM K_2_HPO_4_)—calcofluor staining technique. For it, each half leaf was kept in aniline-blue from 12 to 24 h in dark at room temperature. Then each sample was stained with calcofluor during 30 s approximately, mounted on microscope slides with the adaxial surface uppermost, covered with a glass coverslip, and examined using an epifluorescence microscope at 5×, 10×, 20× (excitation filter 368 nm, dichoric mirror 385 nm, barrier filter 420 nm). Fungal hyphae and conidia could then be observed in fluorescent blue, and both, penetration points and cell walls with callose accumulation, in yellow. The number of penetration points, and the number of cells with callose accumulation in their cell walls (called “cells with callose” from now on) were counted in 30 conidia per sample. In order to evaluate the relative fungal penetration among samples, an estimated ratio between penetrations points and cells with callose (P.points/Callose) has been also considered for analyses. A value of 0.9 was used when no cells with callose were observed.

### 4.3. Fungal Development at Different Temperatures

Plants from several melon accessions were used in order to observe fungal development at different temperatures and at different times post inoculation. Three plants per melon accession and powdery mildew race were used in all cases. Plants of “TGR-1551” and “Bola de Oro” (half leaves taken at 48, 72, 96 and 120 hpi), and of “PMR 45”, “WMR 29”, and NIL21 (half leaves taken at 48 and at 120 hpi), were inoculated with race 2 using an inoculation tower at 70% HR, 16h light/8h dark at 18–22 °C (considered low temperature) and at 25–30 °C (considered high temperature) at the Experimental Station “La Mayora” (Spain). Callose accumulation in penetration points and cell walls, as well as the number of primary hyphae from conidia, were evaluated at 48, 72, 96, and 120 hpi. The number of conidiophores per conidium was recorded at 120 hpi. A *t*-test was employed to detect the significant differences between the two essays carried out at low temperatures (TUM and “La Mayora”) at 72 hpi in “TGR-1551” and “Bola de Oro” for the number of hyphae, penetration points, and cells with callose. This time point (72 hpi) was selected since appropriate discrimination could be established between compatible and incompatible interaction, taking into account our previous essays with “Bola de Oro” and “TGR-1551” inoculated with race 2. Moreover, a *t*-test or an ANOVA was performed to observe the significant differences between high and low temperature in each accession and among accessions for the parameters mentioned above for the experiments carried out in “La Mayora”.

### 4.4. Statistical Analyses

ANOVA analyses were performed to test the significant differences among the melon accessions and among the powdery mildew races examined in all the experiments. When only two races or genotypes were tested, differences were observed with a *t*-test (*p* < 0.05). The statistical analyses were carried out by using the SPSS software [48].

## Figures and Tables

**Figure 1 pathogens-10-00873-f001:**
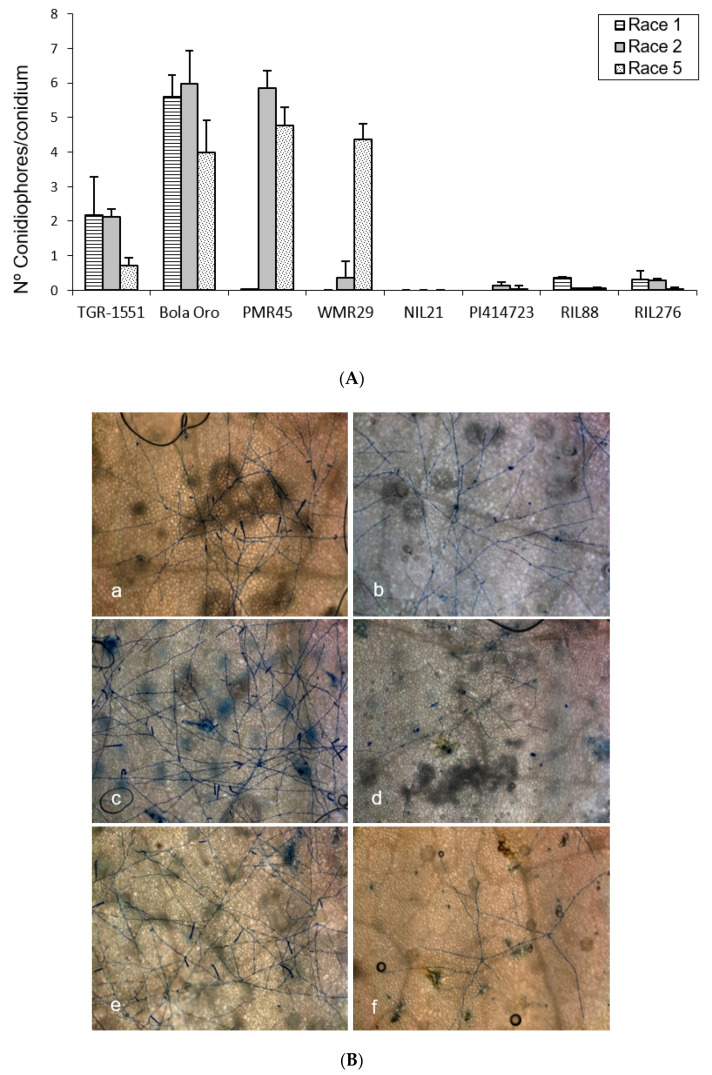
(**A**) Mean number of mature conidiophores per conidium (30 conidia/sample) in second true leaves of three plants of different melon accessions for three powdery mildew races at 120 hpi. No conidiophores were reported for neither the three races in NIL21 nor for race 1 in “PMR 45” and “WMR 29” (not evaluated in “PI 414723”). (**B**) Development of *Podosphaera xanthii* race 5 observed at five days post inoculation in second true leaves from “Bola de Oro” (**a**), “TGR-1551” (**b**), “PMR 45” (**c**), NIL21 (**d**), “WMR 29” (**e**), and “PI 414723” (**f**).

**Figure 2 pathogens-10-00873-f002:**
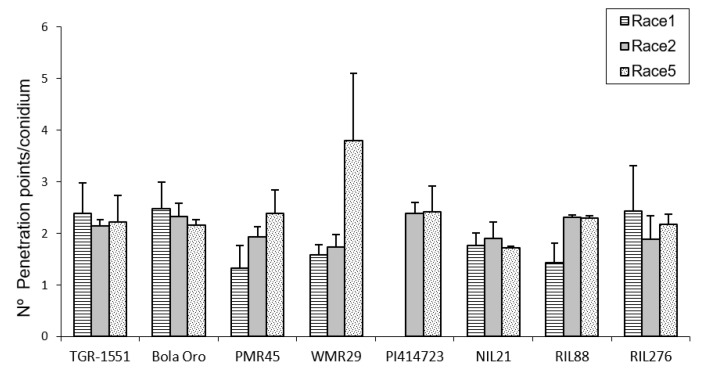
Mean number of penetration points with callose accumulation counted per conidium (30 conidia/sample counted) in second true leaves of three plants of “TGR-1551”, “Bola de Oro”, “PMR 45”, “WMR 29” (one plant), NIL21, RIL88, and RIL276 for races 1, 2, and 5; and in “PI 414723” for races 2 and 5 of *Podosphaera xanthii* at 48 h post inoculation. Race 1 was not evaluated in “PI 414723”.

**Figure 3 pathogens-10-00873-f003:**
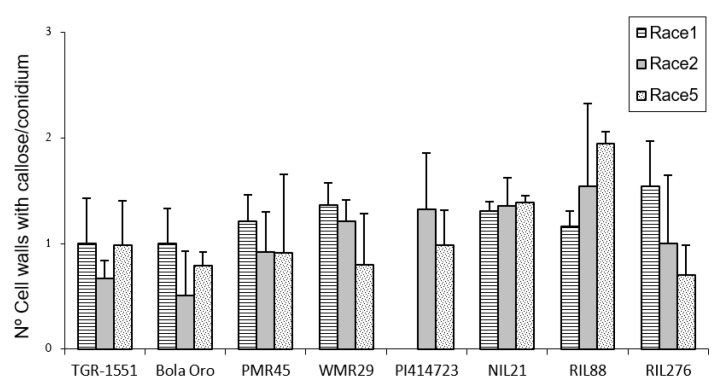
Mean number of cells with callose accumulation in their cell walls (n° cell walls with callose), counted per conidium (30 conidia/sample counted) in second true leaves of three plants of several melon accessions for races 1, 2, and 5 of *Podosphaera xanthii* at 48 h post inoculation. Race 1 was not evaluated in “PI 414723”.

**Figure 4 pathogens-10-00873-f004:**
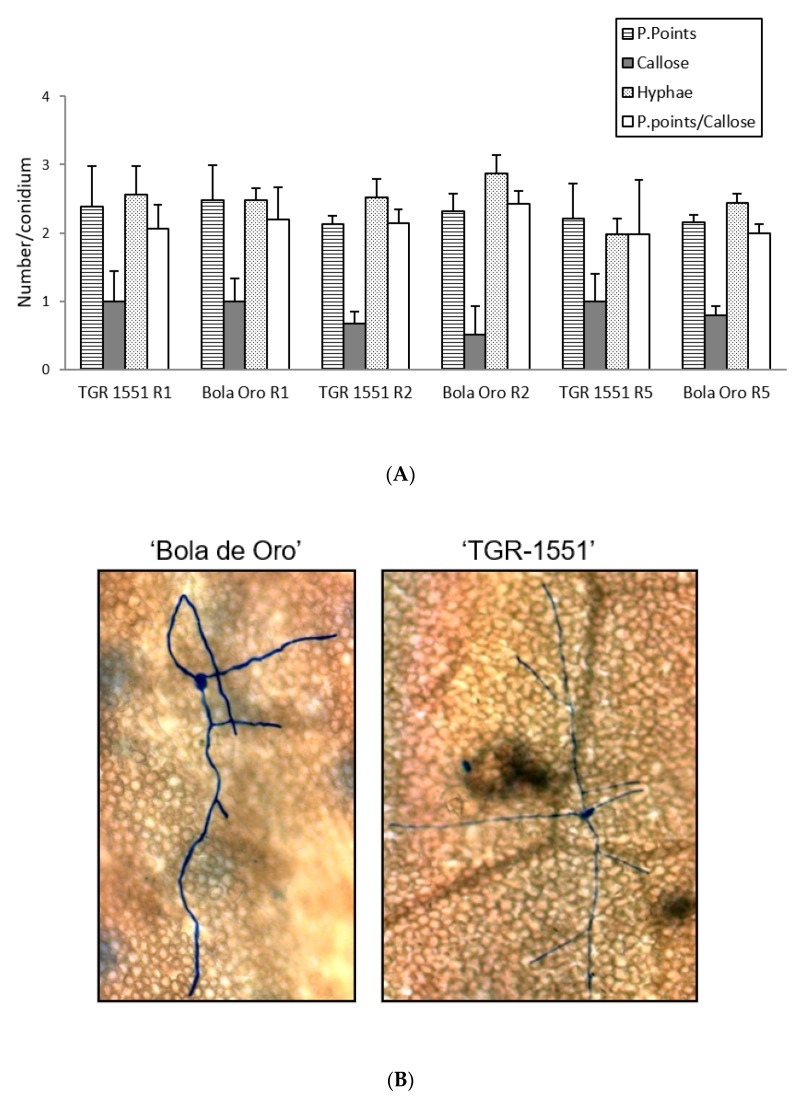
(**A**) Mean number of penetration points with callose accumulation (P. points), cell walls with callose accumulation in their cell walls (Callose), primary hyphae (Hyphae), and the ratio P. points-Callose (P. points/Callose), counted per conidium (30 conidia/sample counted) in second true leaves of three plants of “TGR-1551” and “Bola de Oro” inoculated with races (R) 1, 2, and 5 from *Podosphaera xanthii* at 48 h post inoculation. (**B**) Primary and secondary hyphae of *Podosphaera xanthii* developed from conidia in “Bola de Oro” and “TGR-1551” at 48 hpi (Ink-acetic staining).

**Figure 5 pathogens-10-00873-f005:**
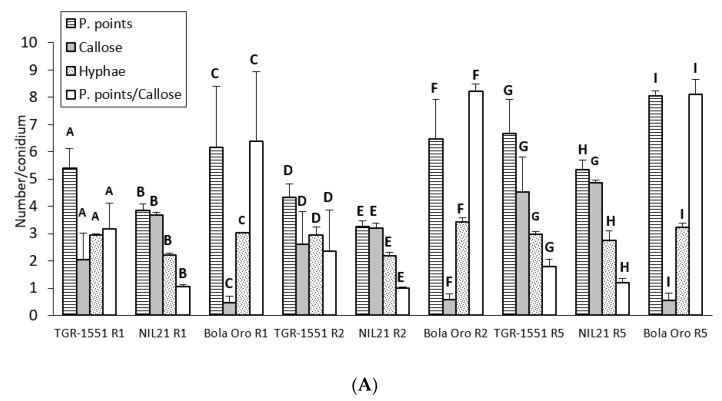
(**A**) Mean number of penetration points with callose accumulation (P. points), cell walls with callose accumulation in their walls (Callose), their ratio (P. points/Callose), and primary hyphae (Hyphae), counted per conidium (30 conidia/sample counted) in second true leaves of three plants of “TGR-1551” and “Bola de Oro” and NIL21 inoculated with races (R) 1, 2, and 5 of *Podosphaera xanthii* at 72 h post inoculation. Differences among the three genotypes for each parameter and race tested are noted from A–C (race 1), D–F (race 2), and G–I (race 5). (**B**) Primary, secondary, and third hyphae of *Podosphaera xanthii* developed from conidia in “Bola de Oro” and “TGR-1551” at 72 hpi (Ink-acetic staining).

**Figure 6 pathogens-10-00873-f006:**
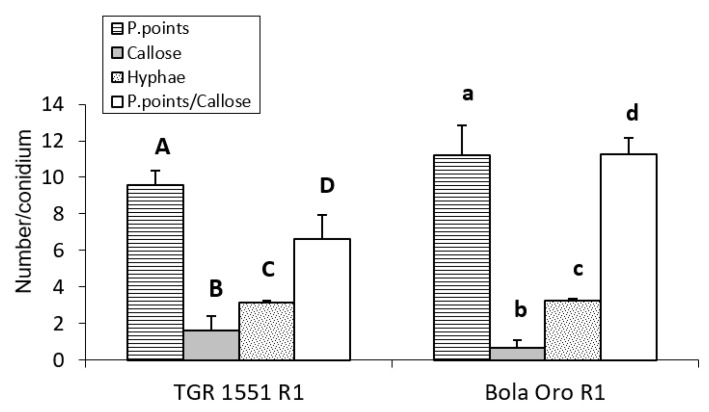
Mean number of primary hyphae (Hyphae) penetration points with callose accumulation (P. points), cell walls with callose accumulation in their cell walls (Callose), and their ratio (P. points/Callose), counted per conidium (30 conidia/sample counted) in second true leaves of three plants of “TGR-1551”, “Bola de Oro”, and NIL21 for race 1 of *Podosphaera xanthii* at 96 h post inoculation. Differences between the genotypes for each parameter tested are noted from A–D, a–d.

**Figure 7 pathogens-10-00873-f007:**
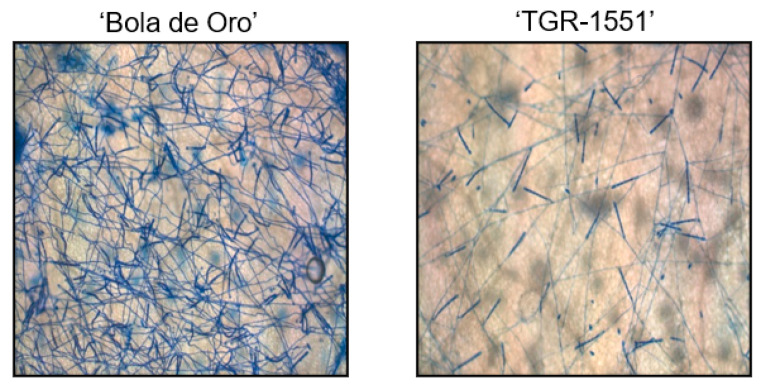
Development of *Podosphaera xanthii* in “Bola de Oro” and “TGR-1551” at 240 hpi (Ink-acetic staining).

**Figure 8 pathogens-10-00873-f008:**
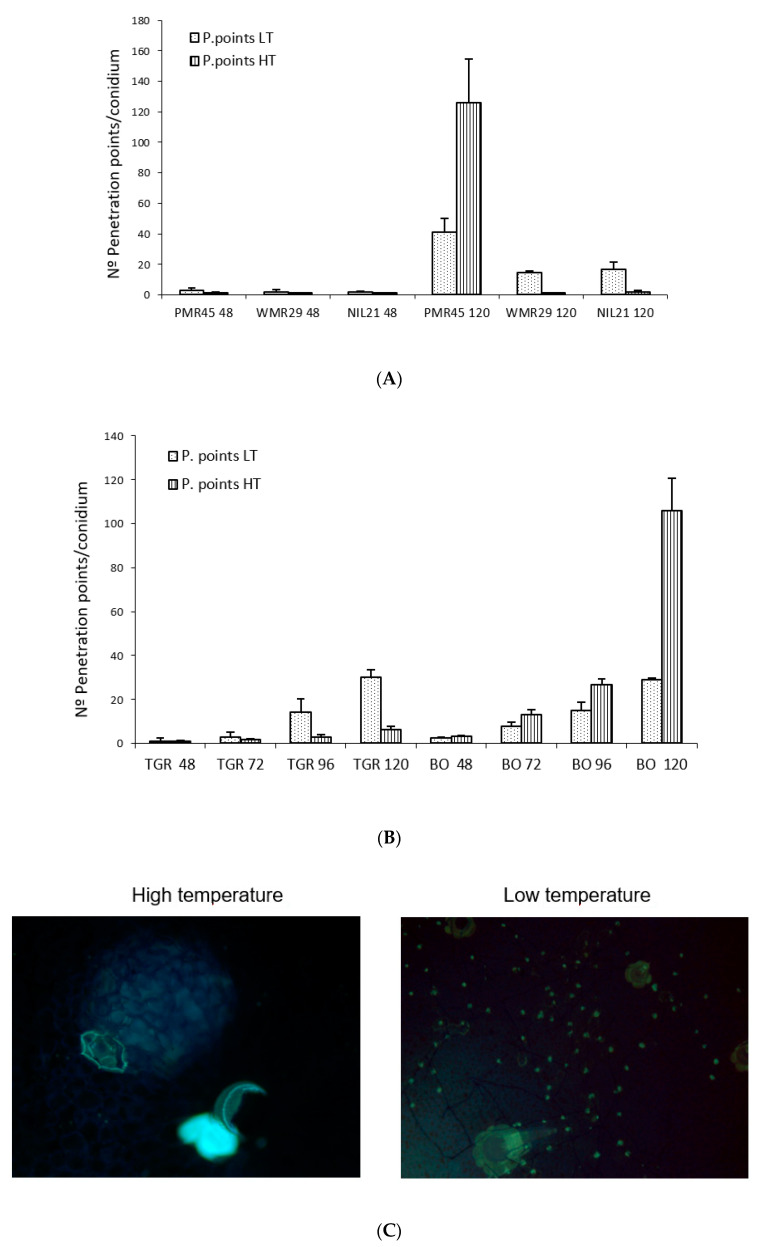
(**A**) Mean number of penetration points per conidium (30 conidia/sample counted) in second true leaves of two to three plants of different melon accessions inoculated with race 2 of *Podosphaera xanthii* at 48 and 120 hpi at low temperature (LT, 18–22 °C) and high temperature (HT, 25–30 °C) (1 plant of “WMR 29” at LT). (**B**) Mean number of penetration points per conidium (30 conidia/sample counted) in second true leaves of two to three plants of “TGR-1551” (TGR) and “Bola de Oro” (BO) in different melon accessions inoculated with race 2 of *Podosphaera xanthii* at 48, 72, 96, and 120 hpi at low temperature (LT, 18–22 °C) and high temperature (HT, 25–30 °C) (1 plant of “TGR-1551” and “Bola de Oro” at 72 and 96 hpi, LT). (**C**) Development of *Podosphaera xanthii* in second true leaves from “TGR-1551” inoculated with race 2 at low and high temperatures (18–22; 25–30 °C) at 96 h post inoculation.

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
