# Peer review of "Fungal Development and Callose Deposition in Compatible and Incompatible Interactions in Melon Infected with Powdery Mildew"

_pathogens, 2021, doi:10.3390/pathogens10070873_

Round 1

Reviewer 1 Report

The authors have already revised very good, my suggestion is accepted.

Author Response

Thank you so much for your message. We are happy for having carried out your revision and suggestions.

Reviewer 2 Report

Title: Fungal development and callose deposition in compatible and incompatible interactions in melon infected with powdery mildew.

Authors: Paola Beraldo-Hoischen, Caroline Hoefle, Ana Isabel Lopez-Sese

Here are some notes that I believe may be helpful in improving the manuscript.

Some are related to small details while some suggest more important changes.

Lines 31-32: “ in coastal zones of the provinces of Malaga and Almeria,” >  maybe not everyone knows that these are two Spanish provinces ...

Lines 310-311 ( and line 570): “Around three plants …..” > in a previous version of the manuscript there was talk of 9 plants for some theses and 3 for others .... to say now that the theses consisted of “around”  3 plants is not correct. If we consider the data of three plants per thesis we must say that they are 3, not about 3

as indicated earlier in the first version of the manuscript:

  1. Line 390: “30 colonies per sample” > how was it ascertained that the colonies were the same size?
  2. Lines 386-403: Are the different data referable to comparable surfaces or did the colonies have different sizes?

The correlation between number of conidia and colony area is not explained. Likewise, it is not stated whether the area of all the colonies was equivalent and therefore irrelevant to comment on the effects of the treatments. This request for clarification was not satisfied. If this clarification is not made, the data relating to the colonies have no meaning and must be removed.

Lines 549-550: “Conidia, primary hyphae, and conidiophores were then coloured in blue >  delete  

Lines 576-578: “Callose accumulation 576 in penetration points and cell walls, as well as the number of primary hyphae, were eval-577 uated at 48, 72, 96 and 120hpi. The number of conidiophores was recorded at 120hpi.” > It is the same problem relating to the size of the areas / colonies where the assessments were made: not described, not specified.

Lines 638 and 641 : “Hordei “ > “hordei “

Line 643: “Canadian journal of botany“ > “Canadian Journal of Botany“

Lines 670 and 671: “H2O2” > format the formula numbers correctly  

Author Response

Thank you so much for your report. Below you will find the revision carried out and the explanations given according to your comments and suggestions.

Question:

Lines 31-32: “ in coastal zones of the provinces of Malaga and Almeria,” >  maybe not everyone knows that these are two Spanish provinces ...

Answer:

Thanks for the warning. It is changed now (line 31).

Question:

Lines 310-311 ( and line 570): “Around three plants …..” > in a previous version of the manuscript there was talk of 9 plants for some theses and 3 for others .... to say now that the theses consisted of “around”  3 plants is not correct. If we consider the data of three plants per thesis we must say that they are 3, not about 3.

Answer:

This issue has been corrected (lines 333; 544; 613-614). Thanks for the warning.

Question:

As indicated earlier in the first version of the manuscript:

  1. Line 390: “30 colonies per sample” > how was it ascertained that the colonies were the same size?
  2. Lines 386-403: Are the different data referable to comparable surfaces or did the colonies have different sizes?

The correlation between number of conidia and colony area is not explained. Likewise, it is not stated whether the area of all the colonies was equivalent and therefore irrelevant to comment on the effects of the treatments. This request for clarification was not satisfied. If this clarification is not made, the data relating to the colonies have no meaning and must be removed.

Answer:

As explained previously, the word ‘colony’ in our manuscript referred to one spore, germinated or not germinated. That means that we estimated all the parameters for each asexual powdery mildew spore (conidium) observed, and the means of all the parameters were calculated for 30 conidia (30 asexual spores) per sample. Therefore, not surface issues are applicable, since they are not bacterial colonies. We included some explanation of this issue in the Material and Methods of our resubmitted manuscript. However, we have now changed the words ‘colony’ for conidium and ‘colonies’ for conidia in the manuscript (text and figures) in order to be more precise and avoid more misunderstandings (see Track Changes: please select ‘All changes’). We hope that this point is now clear and more precise. Thanks so much for your comments.

Question:

Lines 549-550: “Conidia, primary hyphae, and conidiophores were then coloured in blue >  delete  

Answer:

The correction was carried out as suggested.

Question:

Lines 576-578: “Callose accumulation in penetration points and cell walls, as well as the number of primary hyphae, were evaluated at 48, 72, 96 and 120hpi. The number of conidiophores was recorded at 120hpi.” > It is the same problem relating to the size of the areas / colonies where the assessments were made: not described, not specified.

Answer:

This issue about the term ‘colony’ has been solved now in the manuscript, as explained above (third question). Besides, we added the right terms (conidium and conidia) when needed in order to clarify this point (lines 620-621).

Question:

Lines 638 and 641 : “Hordei “ > “hordei “

Answer:

The correction was carried out as suggested (ref. 12 and ref. 13).

Question:

Line 643: “Canadian journal of botany“ > “Canadian Journal of Botany“

Answer:

The correction was carried out as suggested (ref. 14) and check all references listed.

Question:

Lines 670 and 671: “H2O2” > format the formula numbers correctly  

Answer:

The correction was carried out as suggested (ref. 25).

This manuscript is a resubmission of an earlier submission. The following is a list of the peer review reports and author responses from that submission.

Round 1

Reviewer 1 Report

In this manuscript, different melon varieties were inoculated with different races of powdery mildew, and a series of parameters were obtained by observing symptom. It was speculated that these melon varieties contained different disease-resistant genes, and the effects of different temperature and time on the development of conidia and conidia peduncle were also found. I have the following questions:

First, what is the basis of the races of powdery mildew and melon varieties used in this manuscript? Secondly, according to some data, parameters and phenomena in the manuscript, the authors speculated that some melon varieties contained different resistance genes, so whether these resistance genes were known or not? Can you use molecular marker detection to better support the observed phenomena and data? Third, the authors explored the effect of temperature on the development of pathogens, so what is the temperature gradient. Fourth, I haven't seen any pictures and tables in this article, please check them.

Based on the above reasons, this manuscript is not recommended for publish at present.

Reviewer 2 Report

Title: Fungal development and callose deposition in compatible and incompatible interactions in melon infected with powdery mildew.

Authors: Paola Beraldo-Hoischen, Caroline Hoefle, Ana Isabel Lopez-Sese

Here are some notes that I believe may be helpful in improving the manuscript.

Some are related to small details while some suggest more important changes.

Line 84 “cero” > “zero” ?

Lines 82-86 : three classes based on the number od conidiophores:  n=0     1<n> 3      n>3 -         Using a scale with only three classes, differentiated by the presence of only 1-3 conidiophores, could it lead to evaluation errors due to the differentiation of only one conidiophore more or less than the average?

Given that sporulation is easily influenced by various substrate and environmental factors, could the use of this scale be a gamble?

Results: The reference to the table / graph with the results should be highlighted first in the text / sentences. In the current way you read the comment first and only after several lines you find out where the results are reported in detail (e.g. Fig……). This would facilitate faster understanding of the results.

Line 114: “significate differences” > “ significative / significant  differences”

Lines 178-180: “In previous inoculations carried out in our team, ‘TGR-1551’ had never shown such a development of the fungus, therefore another variable might be influencing this fungal development in ‘TGR-1551’.” > Perhaps a more specific comparison of the two experiments could better evaluate any variables that could influence the results. It is also strange, to go without fail on the temperature as a factor to be verified…. Wasn't it known before that the two experiments were different for this factor?

Are you sure that the temperature was the only factor capable of influencing the development of the fungus?

Lines 186-192 / line 355: They seem like planned experiments with a few plants: nine on one side, "a maximum of three" on the other. Also, different leaves are considered replicas but the possible differences between plants have not been considered? why were not the same number of plants used per thesis?

Line 370: “acclimatized chamber” > “climatic chamber”

Line 388: “ink/acetic acid solution” > specify the composition or insert a citation.

Line 390: “30 colonies per sample” > how was it ascertained that the colonies were the same size? Lines 386-403: Are the different data referable to comparable surfaces or did the colonies have different sizes?

Lines 396-397 “epifluorescence microscope” >  specify the filters used for emission and excitation UV light

I would suggest to shorten the part concerning the discussion of the results to increase the ease of acquiring the most important information.

Reviewer 3 Report

I don't feel I have enough information to properly review this manuscript.  The manuscript is difficult to understand, and I strongly recommend that the authors consult an English speaker familiar with the topic to make sure the sentences are more idiomatically constructed with fewer clauses and with correct terminology.  I would also recommend the use of summary tables with clear information on the statistics used, with P values and averages paired with standard error (or standard deviation).

I am rejecting the manuscript with strong encouragement to resubmit; I would be happy to review it once it is a little more understandable.